# Relationship between Socioeconomic Inequalities and Oral Hygiene Indicators in Private and Public Schools in Karachi: An Observational Study

**DOI:** 10.3390/ijerph17238893

**Published:** 2020-11-30

**Authors:** Tamsal Khalid, Syed Sarosh Mahdi, Mariam Khawaja, Raheel Allana, Francesco Amenta

**Affiliations:** 1Department of Community Dentistry, Jinnah Medical & Dental College, Sohail University, Karachi 74800, Pakistan; tamsalkhalid@gmail.com (T.K.); mariamkhawaja12345@gmail.com (M.K.); 2Centre of Clinical Research, Telemedicine and Telepharmacy, School of Medicinal and Health Products Sciences, University of Camerino, Camerino, 62032 Marche, Italy; francesco.amenta@unicam.it; 3Department of Paediatrics & Child Health, Aga Khan University Hospital, Karachi 74800, Pakistan; raheel.allana@aku.edu

**Keywords:** oral health, inequality, marginalization, caries, DMFT

## Abstract

Background: The study investigated the relationship between socioeconomic status and oral hygiene indicators in two schools located in Karachi, Pakistan. Oral hygiene indicators of public and private school children were compared. Private schools cater to children of relatively wealthier families compared to public school, whose attendees are generally children from less affluent backgrounds. The aim of this study was to determine whether socio-economic differences and inequalities have an impact on key oral hygiene indicators. Methodology: Primary data for this research was collected from community school visits conducted by the community dentistry department of Jinnah Medical and Dental and Medical College from January to September 2019. A convenience sample of the two schools, comprising 300 school students was selected. Data was collected using modified World Health Organization (WHO) oral health care forms. A pre-tested/customized dental hygiene form based on WHO forms was created by the research team. This form was used to measure DMFT/dmft scores and key oral hygiene indicators in the sample. Results: A total sample size of 300 school-children affiliated with public and private schools was selected. The children’s age ranged from 2 to 18 years. The mean DMFT scores of private and public-school children were not significantly different (private (1.82) vs. public (1.48)). (*p* = 0.257). The mean of carious teeth was 1.69 in private school children compared to 1.34 in government school children, whereas the mean values of other key indicators of oral hygiene including plaque deposition (*p* = 0.001), dental stains (*p* < 0.001) and bleeding gums/gingivitis (*p* < 0.001), were statistically significant between public and private school children. Conclusion: Oral health inequalities can be reduced with increased awareness and public funding to cater for the oral health needs of children of less affluent families. A dynamic and practical community-oriented program is fundamental for enhancing pediatric oral hygiene status, particularly for children attending government schools.

## 1. Introduction

The oral cavity is a gateway to our bodies. Oral hygiene is the practice of keeping the oral cavity hygienic, free of disease and other associated conditions (e.g., halitosis), through regular teeth brushing (dental hygiene), interdental cleaning and other oral hygiene measures. It is important that oral hygiene measures should be carried out on a daily basis to prevent dental disease and bad breath.

The concept of dental health is not solely the presence of healthy dentition and functional teeth, but the oral cavity encompasses all other hard and soft structures inside the oral cavity including soft tissues, gingiva, surrounding periodontium and hard structures [1]. Edentulism, tooth decay, broken teeth, and mal-positioned teeth not only reduce the quality of life by causing pain and discomfort, but also cause embarrassment in social situations. These oral conditions also make essential tasks like mastication, swallowing and socializing difficult. Dental disease has a negative impact on the study and work schedule of school going children, resulting in precious school hours being wasted yearly, across the world [2]. The most common dental diseases are dental caries (cavities) and gingival diseases, which encompasses gingival and periodontal disease [3]. Dental caries is a complex disease with many causative factors, simultaneously occurring to create an optimum environment for initiation and progression of the caries process. The caries process is dynamic in nature as it involves alternating periods of demineralization of enamel due to low PH, which is a result of bacterial metabolism [4]. Dental caries is a global phenomenon and there is no country or geographical entity with a human population, which does not experience dental caries. The caries process begins as soon as the child develops dentition and the first teeth erupts [5].

In Pakistan, the dental caries disease burden is very high, particularly in children. The prevalence of caries is also higher in rural populations compared to urban centers. The situation is made worse with poor hygiene, oral practices and lack of awareness. Cariogenic dietary products most commonly consumed in Pakistan are carbonated drinks, betel nut, smokeless tobacco and Gutka (a chewing tobacco preparation). The socio-economic and demographic causes of dental caries in Pakistan are often neglected and research mainly focuses on health aspects of the disease, however this is an area that requires more research to ascertain socio-economic variables associated with dental health. Children acquire healthy oral hygiene practices through parental guidance or advice of older family members [5,6]. The vast majority of oral diseases can be prevented by improving the awareness level of general public regarding oral hygiene and general brushing/flossing techniques using community programs. Knowledge regarding oral health is critical for improving indicators of oral health across all segments of the population [7]. According to the National Oral Health Survey of Pakistan, the majority of people do not have access to the basic dental health facilities in both public and private sectors [8]. There are no free dental services offered by the government, and treatments that are offered by private hospitals and clinics are often very expensive, and hence they are not affordable for most patients. Oral hygiene awareness is minimal in children from poor homes, which leads to various dental issues [9].

The aim of this study is to determine whether socio-economic differences and inequalities impact key oral hygiene indicators in children studying in private and public schools in Karachi. Private schools cater to the children of relatively wealthier families as compared to public schools, as tuition fees of private schools is higher as compared to public schools.

## 2. Material and Methods

Primary data for this study was obtained from community school visits undertaken by Jinnah Medical and Dental College from January to September 2019. The research was performed by a team of trained, skilled and certified dental surgeons with the assistance of dental assistants and second year Bachelors of Dentistry (BDS) students who helped in filling out the forms. Senior and junior faculty members of the Community Dentistry Department of Jinnah Medical and Dental College visited one private and public school each in Karachi in two cycles over a period of two weeks. A convenience sample from two schools comprising 300 students was selected. The authors conduced a pilot study to calculate the sample size. The pilot study comprised of 30 students, where the prevalence of carious tooth was observed to be to be around 73%. Based on these findings, sample size analysis was conducted, and sample size was estimated to be around 300. The first cycle of the research was conducted with school children in a private school (*n* = 150), while the second cycle was administered with school children of a public school (*n* = 150). Prior to the school visit and examination, a written consent letter was sent to the parents of the children to seek permission for the dental examination.

All examined children had signed consent of their parents. Children found unwilling for the dental examination on the day of the examination were also exempted.

A pre-tested/customized dental hygiene form based on WHO dental hygiene school form was created by the faculty members of community dentistry department to measure DMFT scores of school children. Children from both schools (*n* = 300) were administered the same examination. The dental examination of the oral cavity of children was carried out in a well-lit, properly-ventilated, designated area. The oral examination was conducted using school chairs and each examination involved a dentist and an assistant. Prior to the dental examination, two senior doctors conducted a 15-min session in each classroom to acquaint the children with the dental examination process. The dental team used dental mirrors and Community Periodontal Index probes to examine the oral cavity based on World Health Organization (WHO) guidelines [10]. Infection control protocols were observed during the examination. Decayed, missing and filled teeth were recorded on the examination form as well as the oral hygiene status and stains. The decayed, missing and filled teeth (DMFT for permanent & dmft for primary dentition) indexes were used to assess the various oral health indicators of the participants; DMFT will be used interchangeably with dmft throughout the manuscript for the sake of uniformity. The DMFT index numerically expresses the caries prevalence by calculating the number of decayed, missing due to caries and filled teeth using the WHO diagnostic criteria. Adding the sum total of these three values gives the total DMFT value and hence the extent of caries or the lack of it [10]. The oral hygiene status of the study participants was evaluated using the Simplified Oral Hygiene index OHI-S. The simplified Oral Hygiene index (OHI) by Green-Vermillion-Hirschman demonstrates the oral hygiene of the patient and expresses the existence of debris/calculus on the surface of the teeth. According to the Greene-Vermillion method, there are three levels of severity of debris/calculus presence. This technique simply enables clinicians to analyze and assess the severity of calculus/debris accumulation and to classify it into three classes from 0 to 6 [11]. The observed level of oral hygiene correlating with oral hygiene index score 0–1.2 was classified as good, 1.3–3.0 as designated as fair whereas 3.1–6.0 was marked as poor [12]. Gingival health was measured using gingival index and Lobene stain index was used for measuring the severity of stains. The presence or absence of dental plaque deposition was examined with the help of plaque index. Demographic data were also collected.

### 2.1. Statistical Analysis

The dataset was analyzed using SSPS Version 20 software package (IBM Corporation, SPSS Inc., Chicago, IL, USA). Descriptive statistics with frequency, mean and standard deviation were computed. The Chi-Square test was conducted to investigate the relationship between DMFT and other factors including age, gender and socio-economic status. Inferential statistics was also utilized by administering Levene test to assess the equality of variance for various variables of the two groups. The two-tailed t-test was used to ascertain whether the differences in the means of the examined variables were significant, with *p*-values (level of significance) set at 0.05.

### 2.2. Ethics Approval and Consent to Participate

The ethical approval was obtained from the Ethical Review Committee of Jinnah Medical & Dental College with protocol #000011/20 Informed consent was taken from parents of all school children before the examination. This was a voluntary dental examination and records were kept anonymous.

## 3. Results

A total of 300 school children from public and private sector schools were selected. Equal number of students from private (*n* = 150), and Public schools (*n* = 150) were sampled. The pupil’s’ ages ranged from 2 to 18 years. Mean age was 7.5 ± 6 years. Table 1 shows the frequency and percentages of the pupils’ oral hygiene status, dental stains, bleeding gums and plaque deposition. Good oral hygiene was observed more frequently in private school children (55.3%) compared to of public-school children (38.7%). Dental stains were more often present in public school children (54.7%) compared to private public-school children (13.3%). The proportion of children with bleeding gums (gingivitis) was higher in public school children (37.3%) compared to private school children (20.0%). Plaque deposition was recorded in 66.7% of public sector school children and 40.0% in private sector school children respectively (Table 1). We found significant differences in the proportions of public and private school children with plaque deposition (*p* < 0.001), dental stains (*p* < 0.001) and bleeding gums (*p* < 0.001), (Table 2). The mean DMFT scores of private and public-school children were not significantly different (private (1.82) vs. public (1.48)). (*p* = 0.257). The mean of carious teeth was 1.69 in private school children compared to 1.34 in government school children (*t*-test = 1.118; 95% Confidence Interval 0.235–0.855; Table 2 and Table 3).

## 4. Discussion

Oral health is an important indicator of health. Across the world, oral health indicators have been improving constantly over the last few decades, but that improvement has not been uniform and dental health inequalities are growing in developing countries due to lack of awareness and resources [13]. It is well known that lower socioeconomic status is associated with poorer health [14]. In Pakistan oral health has remained an immense cause of concern. Broad socioeconomic and ethnic differences in both the prevalence and severity of oral diseases have been observed [15,16]. Despite improvements in the oral health situation in developed countries over the last few decades, oral health services are still considered a luxury in developing countries like Pakistan, and awareness levels remain low [17]. In this regard dental diseases, oral health and overall oral hygiene should be very high on the research agenda. This study was conducted to shed light on oral health inequalities that exist in Pakistan owing to widening income and social inequalities.

This study was carried out on children in Karachi with age ranges from 2 years to 18 years. Age has significant bearing on oral health and studies have found that children’s age greatly influences their oral health status. According to American Dental Association (ADA), early childhood caries, (ECC) occurs in children between birth and 71 months of age [18]. A study conducted in Lahore, Pakistan found that prevalence of caries was 33.3% for 3-year old children, 47.6% for 4-year-old children and 75% for 5-year-old children [19]. Peretz et al. [20] conducted a longitudinal study to assess the progression of early childhood caries. children between 3–5 years of age were included and their follow up visits were scheduled after they turned 7. The study findings demonstrate that children with ECC had 1.15 ± 0.97 additional areas affected per year, while caries free children had an increase of only 0.41 ± 0.60 per year, and children with posterior caries had an increase of 0.74 ± 0.64 per year. In the United States, 20% of the children aged 5–11 years have at least one tooth with untreated decay and there are marked socioeconomic disparities in dental caries experiences. Socio-economic disparities are also evident in adolescents aged 5–19, where 25% of children from low income families have untreated tooth decay as compared to 12% of children from higher income families [21]. A few studies suggest that 5 year-olds are more susceptible to caries compared to the 10–14 year-olds, as deciduous teeth contains less enamel thickness due to fact that calcium content in primary teeth is thinner compared to permanent teeth, and also because younger children have poorer oral hygiene, even with parental aid [22]. Tooth brushing for younger children remains irregular and sugar intake remains high [23]. Tooth morphology of younger children also makes them more susceptible to caries [24] and consumption of artificially sweetened sugar products also places children at higher risk of developing dental caries [25]. Health promotion directed at pregnant women, new mothers and primary caregivers should raise questions about the popular risk factors of ECC, by stressing upon WHO guidelines for breast-feeding up to six months of age [26].

This study revealed no significant difference in mean DMFT scores of public and private school children, but there were statistically significant differences in other indicators of oral health including stains, plaque deposition and bleeding gums. Public school students fared badly in all three of these oral health indicators compared to their counterparts in private schools. This is significant evidence that socioeconomic status has a significant bearing on key oral hygiene indicators, as pupils of public schools fared badly on these indicators, compared to private school pupils [27]. These findings are consistent with other studies carried out on the same subjects in the world [28]. Though dental stains are not a disease and do not cause any disease, we chose to include staining in this study because many intrinsic stains can be caused by various underlying diseases such as thalassemia, sickle cell anemia, and porphyria. On the contrary, external stains are caused by poor oral hygiene/inadequate tooth brushing and these extrinsic stains are early signs of tooth decay [29,30]. Stains were used as an additional measure to understand the general oral hygiene status of the children under study. The oral examination by the examiners revealed that most of these stains were mostly extrinsic in nature, hence signifying poor oral hygiene rather than any systemic factors.

This study also revealed that the mean of carious teeth was 1.69 in private school children and 1.34 in government school children and that these differences were not statistically significant. These results were higher than the mean of decayed teeth in other studies conducted in Pakistan (0.87 ± 1.1) [31]. It is a matter of debate, why public school students were not observed to have more carious teeth compared to private school students, despite performing poorly on all other oral health indicators, like stains, plaque deposition and bleeding gums. One explanation is the fact that children from well off families get more pocket money from their parents and they are more likely to spend that money on carbonated drinks, chocolates and confectionary products, particularly during school lunch breaks. Consuming these products, particularly during school lunch breaks, leaves them vulnerable to caries formation as the next tooth brushing session would not take place for several hours. Research carried out in China on oral hygiene of school children found that the prevalence of dental caries amongst school children aged 5–7 was 76.6%, likewise a similar study conducted on school children aged 6–10 in Mexico, found that the prevalence of dental caries was 65.5% [32]. Similarly, prevalence of dental caries in our study was 51.6%. The mean of missing teeth in our study was 0.73 and 0.147 in private and public school children respectively, which was higher than that of same age group children studied in India and Pakistan [31]. Similarly the frequency of filled teeth was also high in our study, when compared to the results of other studies in Pakistan [31]. The reason of these differences could be regional variations in dietary habits, Karachi is a cosmopolitan urban sprawl and hence children are more exposed to advertisement campaigns of confectionary products/cola drinks/chocolates more than their rural and smaller city counterparts. The mean DMFT in our study was 1.82 in private school children and 1.48 in public school children, which was statistically similar to the study of Bardal et al. in Brazil, among 7–12 years old school children, where the mean DMFT score was also found to be 1.82 [33]. However it was lower than that of Indian children (1.9) and higher than that of another study conducted in a poor locality of the Pakistani city of Lahore (1.0) [34]. Our study had equal sample size from children of upper middle class families as well as poorer families (150 each), that is why we believe the DMFT scores are different from studies conducted on children of entirely poor households or children coming from well off backgrounds. Moreover, when our mean DMFT score was compared to a study from Chennai, India, the difference in mean DMFT scores was more than twice (DMFT = 3.94) the mean DMFT observed in our study. Which indicates that that oral hygiene habits/attitudes vary greatly between various communities living in the subcontinent. Similarly, a previous study from Pakistan [35] reported much higher DMFT (3.7), than that reported in the current study. The previous study was carried out a decade ago and since then oral hygiene attitudes/knowledge have generally improved as referenced previously. We believe that the advent of social media has also increased peer pressure on children and their families to improve oral health and hygiene. Our study also concluded that there was a significant difference in dental stain proportions in private and public-school children, 13.3% and 54.7% respectively and this a unique element of this study, as this aspect of oral health has not been studied locally. This difference is understandable as well-off families are more likely to invest on appearances and aesthetics. A study conducted on school going children in Saudi Arabia found that the mean DMFT score among children aged 5–12 year old was 3.85 [36].

The proportion of bleeding gums was observed to be more prevalent in public school children (37.3%), than that of private school children (20.7%), which was slightly more than that observed in a similar study of school going children in Iraq, Baghdad (33.4%) [37].

Oral hygiene status was good in 47.0% (private = 55.3%, Govt. = 38.7%), Fair in 35.3% (private = 28%, Govt. = 42.7%) and bad in 17.7% (private = 16.7%, Govt. = 18.7%) [38]. Our findings are inferior to the results obtained by Bashirian S et al., and better than that of another study conducted in Lahore in school going children [39].

These result show inadequacy in basic oral hygiene protocols among the school children, particularly in public school children. Hence, it is critical to formulate oral health promotion policies directed specifically towards the public sector schools and institutes. It is also fundamental to raise the standards of dental care by educating students about preventive measures and application of cost-effective interventions. Some limitations were also considered in this study.

Our study only examined two schools. Therefore, the results cannot be considered national representative figures.As the sample size was moderate, hence our results cannot be generalized. A larger national study might also make the differences in DMFT levels between private and public-school students statistically significant. Future studies are required with a larger sample size.Some important aspects were not examined in this study that might have an impact on oral health. These include nutritional habits, knowledge and opinions about dental health and parent’s attitude towards oral health.

## 5. Conclusions

Our research findings indicate that the general oral health status of private school children was better compared to children of public schools, which suggests the influence of socio-economic status on dental health of children. There is a dearth of research on oral health of school children in Pakistan and more studies should now be conducted nationwide, in order to understand the oral hygiene status of school going children, especially in the public sector, which can be then used to guide policy making with regards to oral health of school children. Furthermore, prevention of early childhood caries should be incorporated into current primary health care programs, particularly those for maternal and child health, as well as school-based community dental programs. Primary caregivers should be trained to provide adequate teeth brushing technique, from the time of first primary tooth eruption, accompanied by early diagnosis of early caries. The management of the ECC should be part of the training for all health practitioners, who facilitate the health of infants and children in society [26]. We conclude that a dynamic and cost effective community based oral health program is fundamental for improving the oral hygiene situation for school going children of both the private and public sector. More focus is required on public school children due to the disadvantageous socio-economic status of these children. Dental institutes with the cooperation of local governments and NGO’s should organize regular preventive programs in public schools, particularly in schools located in poor suburbs and slum areas to address the growing burden of oral diseases.

Importance for pediatric dentistry:
The study could serve as a model for a future larger-scale assessment of the oral health of schoolchildren in Pakistan.Preventive community programs in schools could significantly reduce caries burden in the most vulnerable children.School-based caries interventions such as application of fluoride and sealants could be used as cost-effective interventions in schools to reduce caries burden [40].


## Figures and Tables

**Table 1 ijerph-17-08893-t001:** Hygiene Status, Dental Stains, Bleeding Gums and Plaque Deposition.

Variables	Categories	School		*p* Value
		Private	Govt.	
Hygiene Status	Good	83(55.3%)	58(38.7%)	0.01
	Fair	42(28.0%)	64(42.7%)	
	Bad	25(16.7%)	28(18.7%)	
Stains	No	130(86.7%)	68(45.3%)	0.000
	Yes	20(13.3%)	82(54.7%)	
Bleeding Gums	No	119(79.3%)	94(62.7%)	0.001
	Yes	31(20.7%)	56(37.3%)	
Plaque Deposition	No	90(60.0%)	50(33.3%)	0.000
	YES	60(40.0%)	100(66.7%)	

**Table 2 ijerph-17-08893-t002:** Showing independent samples test and 95% Confidence Interval.

Independent Samples Test										
		Levene’s Test for Equality of Variances		T-Test for Equality of Means						
		F	Sig.	t	df	Sig. (2-Tailed)	Mean Difference	Std. Error Difference	95% Confidence Interval of the Difference	
									Lower	Upper
Age	Equal variances assumed	56.428	0.000	−14.203	298	0.000	−4.9667	0.3497	−5.6548	−4.2785
	Equal variances not assumed			−14.203	263.189	0.000	−4.9667	0.3497	−5.6552	−4.2781
Carious Teeth	Equal variances assumed	19.359	0.000	1.283	298	0.201	0.3467	0.2702	−0.1851	0.8784
	Equal variances not assumed			1.283	229.928	0.201	0.3467	0.2702	−0.1857	0.8791
Missing	Equal variances assumed	8.231	0.004	−1.567	298	0.118	−0.0733	0.0468	−0.1654	0.0188
	Equal variances not assumed			−1.567	295.416	0.118	−0.0733	0.0468	−0.1655	0.0188
Filled	Equal variances assumed	1.395	0.238	0.584	298	0.560	0.0133	0.0228	−0.0316	0.0583
	Equal variances not assumed			0.584	268.445	0.560	0.0133	0.0228	−0.0316	0.0583
DMFT	Equal variances assumed	16.623	0.000	1.136	298	0.257	0.313	0.276	−0.230	0.856
	Equal variances not assumed			1.136	236.664	0.257	0.313	0.276	−0.230	0.857

**Table 3 ijerph-17-08893-t003:** Differences in DMFT Between Private and Public-School Children.

	t	Sig. (2-Tailed)	Mean Difference	95% Confidence Interval of the Difference	
				Lower	Upper
Age	−14.203	0.000	−4.9667	−5.6548	−4.2785
Carious Teeth	1.283	0.201	0.3467	−0.1851	0.8784
Missing	−1.567	0.118	−0.0733	−0.1654	0.0188
Filled	0.584	0.560	0.0133	−0.0316	0.0583
DMFT	1.136	0.257	0.313	−0.230	0.856

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
