# Peer review of "Relationship between Socioeconomic Inequalities and Oral Hygiene Indicators in Private and Public Schools in Karachi: An Observational Study"

_ijerph, 2020, doi:10.3390/ijerph17238893_

Round 1
Reviewer 1 Report
I have made three minor corrections to the text, and it is now acceptable. However, both reference 25 and 28 need revising because they are currently written in capital letters. Once that is done, the paper can proceed to publication.
Author Response
Reviewer 1
Comment 1:
I have made three minor corrections to the text, and it is now acceptable. However, both reference 25 and 28 need revising because they are currently written in capital letters. Once that is done, the paper can proceed to publication.
Response: The authors thank the reviewer for the encouraging comment and also help with English editing. The references have been revised according to the suggestions offered by the esteemed reviewer. We thank you for your gracious review.
Reviewer 2 Report
The authors revised the manuscript, but I really do not see many changes. For example, the authors added the sentences, "Age has an influence on the oral condition of the children. 20% of the children aged 5-11 years in the United States have at least one decayed tooth which is untreated in the United States.... One study conducted in Tanzania revealed that frequent intake of factory-made sugar products at the age of 1-2 years was correlated with higher risk of developing dental caries 22." I think that this is just an introduction of references and there is no consideration. Moreover, they cited convenient references develop the story based on the current results. So, this manuscript lacks consistency form Introduction section to Conclusion section.
Author Response
Reviewer 2
Comment 1:
The authors revised the manuscript, but I really do not see many changes. For example, the authors added the sentences, "Age has an influence on the oral condition of the children. 20% of the children aged 5-11 years in the United States have at least one decayed tooth which is untreated in the United States.... One study conducted in Tanzania revealed that frequent intake of factory-made sugar products at the age of 1-2 years was correlated with higher risk of developing dental caries 22." I think that this is just an introduction of references and there is no consideration. Moreover, they cited convenient references develop the story based on the current results. So, this manuscript lacks consistency form Introduction section to Conclusion section.
Response: The authors thank the reviewer for this incisive comment. This is our clarification and further action to the query. Detailed explanation on the role of age and oral health has now been offered in addition to the previous changes we made with regards to the same query in the discussion section. The evidence for early child hood caries and other forms of dental caries /decay is almost universal in children and we have tried to add all key references to this effect. We have added both local and international sources to make the argument more concise. The following section has been added in the discussion section
- According to American Dental Association, Early childhood caries (ECC) occurs in children between birth and 71 months of age and this greatly influences their overall quality of life 1. A study conducted in Lahore, Pakistan by Sufia S et al found that prevalence of caries was 33.3 % for 3-year-old children, 47.6 % for 4-year-old children and 75 % for 5-year-old children. This showed that if early caries left untreated at an early age it can progress which is evident from this study findings 2. Peretz et al 3 conducted a longitudinal study to assess the progression of early child hood caries. Children age between 3-5 years of age were included and their follow up visits were at atleast 7 years since their initial visit. The study findings demonstrate that Children with ECC had 1.15±0.97 additional areas affected per year, while caries free children had an increase of 0.41±0.60 per year, and children with posterior caries had an increase of just 0.74±0.64 per year. Thus Children with ECC could have a high chance of developing potential carious lesions relative to caries-free children. In the United States, 20% of the children aged 5-11 years have at least one decayed tooth which is untreated. The socio-economic disparity is also evident in adolescents aged 5-19, where 25% of children from low income families have untreated tooth decay as compared to 12% of children from higher income brackets 4. Few studies suggests that 5 years old are more susceptible to caries compared to the age group of 10-14 as deciduous teeth contains less enamel thickness as the calcium content in primary teeth is less compared to permanent teeth and also due to the fact younger children are less keen on tooth brushing, even in the presence of parents. Tooth brushing for younger children remains irregular and sugar intake remains high. Tooth morphology of younger children also makes them more susceptible to caries 5, 6, 7. One study conducted in Tanzania also revealed that frequent intake of factory-made sugar products at the age of 1-2 years was correlated with higher risk of developing dental caries 8.
Health promotion directed at pregnant women, new mothers and primary caregivers should raise questions about the popular risk factors of ECC by stressing WHO guidelines for breast-feeding up to six months of age. No additional sugars for supplemental feeding up to two years and, consequently, limited free sugar consumption in compliance with the WHO guidelines 9.
- The authors have also revised the methodology portion and the research design has been elaborated more rigorously. We have also edited the manuscript for typographical errors and English language. We hope that these changes will satisfy the reviewer.
References
- Statement on Early Childhood Caries. Ada.org. https://www.ada.org/en/about-the-ada/ada-positions-policies-and-statements/statement-on-early-childhood-caries. Published 2020. Accessed October 6, 2020.
- Sufia S, Chaudhry S, Izhar F, Syed A, Mirza BA, Khan AA. Dental caries experience in preschool children: is it related to a child's place of residence and family income? Oral Health Prev Dent. 2011;9(4):375-9
- Peretz B, Ram D, Azo E, Efrat Y. Preschool caries as an indicator of future caries: a longitudinal study. Pediatr Dent. 2003 Mar-Apr;25(2):114-8.
- Dye BA, Xianfen L, Beltrán-Aguilar ED. Selected Oral Health Indicators in the United States 2005–2008. NCHS Data Brief, no. 96. Hyattsville, MD: National Center for Health Statistics, Centers for Disease Control and Prevention; 2012
- Malvania EA, Ajithkrishnan CG, Thanveer K, Hongal S. Prevalence of dental caries and treatment needs among 12 years old school going children in Vandodara city, Gujrat, India. A cross sectional study. Indian J Oral Sci. 2014; 5:3-9.
- Peedikayil FC, Kottayi S, Kenchamba V, Jumana MK. Dental caries prevalence and treatment needs of school going children in Kannur disrict, Kerala. Journal of RDS. 2013; 4: 51-53.
- Dukic W, Delija, Lulic Dukic O. Caries prevalence among schoolchildren in Zagreb, Croatia. Croat Med J.2011;52:665-671.
- Mwakayoka H, Masalu JR, Namakuka Kikwilu E. Dental Caries and Associated Factors in Children Aged 2-4 Years Old in Mbeya City, Tanzania. J Dent (Shiraz). 2017;18(2):104-111.
- WHO Global Consultation on Public Health Intervention against Early Childhood Caries. World Health Organization. https://www.who.int/oral_health/publications/global-consultation-intervention-against-early-childhood-caries/en/. Published 2020. Accessed October 7, 2020.
Reviewer 3 Report
The revised manuscript properly reflected the reviewer's opinions. The specific interventions should be needed to improve the oral health of the vulnerable children. This article will contribute to the building strategies to reduce the oral health inequalities among children caused from the ineqalities of socioeconomic status. English language and style are minor check required.
Author Response
Reviewer 3
The revised manuscript properly reflected the reviewer's opinions. The specific interventions should be needed to improve the oral health of the vulnerable children. This article will contribute to the building strategies to reduce the oral health inequalities among children caused from the inequalities of socioeconomic status. English language and style are minor check required.
Comment 3:
Thank you for your valuable and encouraging comments. English language/typographical errors have been accounted for and minor checks have been completed as per instructions.
This manuscript is a resubmission of an earlier submission. The following is a list of the peer review reports and author responses from that submission.
Round 1
Reviewer 1 Report
This is a worthwhile paper that merits publication. However, there are several errors in the original, so much so that I have provided my edited version (attached) with improved English. For example, it is not necessary to express the term as DMFT/dmft. These two versions differ only in whether or not capital letters are used, so I have changed it to DMFT in every case.
Also, try not to use slang, e.g. referring to children as kids.
I do not agree that ART has gone out of favour in the last few decades. It is only about 20 years old as a technique, and is clearly "in favour" in may countries, as shown by the current literature. Consequently, I have edited this sentence.
Finally, there are major mistakes with nearly every reference. Mostly, the title of the journal does not appear in the proper place (i.e. after the title of the paper) or in the proper format. This remains to be corrected by the authors.
Despite this concerns about the presentation, I consider this to be a useful contribution and one that will merit publication once the necessary corrections have been made.

Reviewer 2 Report
This article explains the relationship between socioeconomic inequalities and oral hygiene indicators in private and public schools in Karachi, Pakistan. The authors have provided information about the differences between the children of private and public schools. Although the point of view of the authors is unique and interesting, I believe that this manuscript does not provide sufficient data and theoretical considerations of the socioeconomic inequalities and lacks new insights.
Few comments to the authors are as follows:
1) Please provide a suitable explanation as to why private schools cater to the children of relatively wealthier families as compared to public schools.
2) Explain the aim of the study clearly.
3) The measurement of the bleeding gums, stains, abnormality in teeth, and presence or absence of deposition of a plaque was not clear.
4) Did the participants receive an explanation about this study before the dental examination? Did the participants get an opportunity to refuse to provide his/her data?
5) The student’s age ranges from 2 to 18 years, which seems to be a broad range. The authors should consider what kind of influence age has on the oral condition.
6) Figures 1–3 are unnecessary. The data from these figures should be included in Table 1.
7) Why did the public school students have fewer caries teeth than the private school students despite more stains, plaque deposition, and bleeding gums?
8) The sentence “This study also revealed that the mean of carious teeth was 1.69 in private school going children and 1.34 in government school going children,” was mentioned by the authors; however, these data were not included in the Results section.
Reviewer 3 Report
- Age of subjects was described at only abstract and it was not described at Materials and Methods.
- Oral examiners were only described as "senior and junior facuty members of community dentistry department". The number of examiners and the calibration process to apply same criteria to diagnosis dental caries status, hygiene status, stains, bleeding gums and plaque deposition should be described.
- The stain is not disease and do not cause diseasae. The reason to include the examination of stain should be described.
- The color of dental plaque is so similar to color of teeth. Please describe the cheking method of dental plaque. Did examiners check the natural status of plaque or check the plaque after disclosing stain?
- The criteria of the classification of good, fair, bad on the hygiene status should be described.
- Authors provided the figures on the proportion of oral hygiene status (Figure 1), plaque deposition (Figure 2) and stains (Figure 3). However, the figures are not clear to indicate the status of which school between private and government school. The figures to compare the status of private and government school on the variables are nessary to readers.